# Efficacy of Intravenous Elosulfase Alfa for Mucopolysaccharidosis Type IVA: A Systematic Review and Meta-Analysis

**DOI:** 10.3390/jpm12081338

**Published:** 2022-08-20

**Authors:** Chung-Lin Lee, Chih-Kuang Chuang, Yu-Min Syu, Huei-Ching Chiu, Yuan-Rong Tu, Yun-Ting Lo, Ya-Hui Chang, Hsiang-Yu Lin, Shuan-Pei Lin

**Affiliations:** 1Department of Pediatrics, MacKay Memorial Hospital, Taipei 10449, Taiwan; 2Institute of Clinical Medicine, National Yang-Ming Chiao-Tung University, Taipei 11221, Taiwan; 3Department of Rare Disease Center, MacKay Memorial Hospital, Taipei 10449, Taiwan; 4Department of Medicine, Mackay Medical College, New Taipei City 25245, Taiwan; 5MacKay Junior College of Medicine, Nursing and Management, New Taipei City 25245, Taiwan; 6Division of Genetics and Metabolism, Department of Medical Research, MacKay Memorial Hospital, Taipei 10449, Taiwan; 7College of Medicine, Fu-Jen Catholic University, New Taipei City 24205, Taiwan; 8Department of Pediatrics, Far Eastern Memorial Hospital, New Taipei City 22021, Taiwan; 9Department of Medical Research, China Medical University Hospital, China Medical University, Taichung 40402, Taiwan; 10Department of Infant and Child Care, National Taipei University of Nursing and Health Sciences, Taipei 11219, Taiwan

**Keywords:** mucopolysaccharidosis type IVA, Morquio A, enzyme replacement therapy, elosulfase alfa, meta-analysis

## Abstract

Mucopolysaccharidosis type IVA (MPS IVA or Morquio A), a lysosomal storage disease with an autosomal recessive inherited pattern, is induced by *GALNS* gene mutations causing deficiency in N-acetylgalactosamine-6-sulfatase activity (GALNS; EC 3.1.6.4). Currently, intravenous (IV) enzyme replacement therapy (ERT) with elosulfase alfa is employed for treating MPS IVA patients. A systematic literature review was conducted to evaluate the efficacy and safety of IV elosulfase alfa for MPS IVA by searching the National Center for Biotechnology Information, U.S. National Library of Medicine National Institutes of Health (PubMed), Excerpta Medica dataBASE, and Cochrane Library databases, limited to clinical trials. Four cohort studies and two randomized controlled trials, with a total of 550 participants (327 on ERT treatment versus 223 on placebo treatment), satisfied the inclusion criteria. Pooled analysis of proportions and confidence intervals were also utilized to systematically review clinical cohort studies and trials. Per the pooled proportions analysis, the difference in means of urinary keratan sulfate (uKS), 6-min walk test, 3-min stair climb test, self-care MPS-Health Assessment Questionnaire, caregiver assistance and mobility, forced vital capacity, the first second of forced expiration, and maximal voluntary ventilation between the ERT and placebo treatment groups were −0.260, −0.102, −0.182, −0.360, −0.408, −0.587, −0.293, −0.311, and −0.213, respectively. Based on the currently available data, our meta-analysis showed that there is uKS, physical performance, quality of life, and respiratory function improvements with ERT in MPS IVA patients. It is optimal to start ERT after diagnosis.

## 1. Introduction

Mucopolysaccharidosis IVA (MPS IVA, Morquio A syndrome, Morquio–Brailsford syndrome, OMIM 253000), a lysosomal storage disease exhibiting an autosomal recessive inherited pattern, is caused by N-acetylgalactosamine-6-sulfatase enzyme deficiency (GALNS; EC 3.1.6.4) due to *GALNS* gene mutations located on chromosome 16q24 [1]. This causes glycosaminoglycans (GAGs) accumulation of keratan sulfate (KS) and chondroitin-6-sulfate (C6S) in the tissues, bones, and major organs [2,3,4,5,6].

The prevalence of MPS IVA in Denmark, the UK, Australia, and Malaysia is 1/323,000, 1/599,000, 1/926,000, and 1/1,872,000, respectively. The birth prevalence rate of MPS IVA ranges from 1/71,000 to 1/500,000 in the United Arab Emirates and Japan, respectively [7,8]. Patients with Morquio A syndrome seem healthy at birth but subsequently develop multiorgan signs and symptoms, and may be diagnosed by the age of 6 months.

Patients with MPS IVA manifest features such as waddling gait, abnormal skeletal development, genu valgum (knock knees), bell-shaped chest, joint hypermobility and laxity, spinal deformities, large elbows and wrists, and short stature for their age with a short neck. Moreover, mild hepatosplenomegaly, hearing impairment, respiratory compromise, abnormal heart development, corneal clouding, and deficient tooth enamel may be present [9,10,11,12,13,14]. Furthermore, due to macroglossia, temporomandibular joint stiffness, abnormal laryngeal anatomy, trachea deformity, and glottic narrowing predisposes the patients toward difficult and failed intubations, thereby putting such patients at a greater anesthetic risk [15]. Compared with other MPS types, Morquio A has no brain involvement nor does it cause significant cognitive developmental impairments [16,17].

Recently, patients with MPS IVA have been receiving supportive symptomatic care, such as adenotonsillectomy for obstructive sleep apnea, keratoplasty for corneal clouding, and hearing aids for sensorineural hearing loss. Furthermore, skeletal abnormalities in some patients require aggressive supportive and symptomatic operative and non-operative interventions. Additionally, there have been reports of beneficial outcomes in MPS from hematopoietic stem cell transplantation (HSCT) [18]. However, due to a limited number of MPS IVA patients who undergo HSCT, data for the mortality rate regarding HSCT for MPS IVA have not been reported [19].

Enzyme replacement therapy (ERT) (elosulfase alfa; Vimizim^®^, Biomarin, Novato, CA, USA) has become the standard treatment for MPS IVA. ERT has several limitations, such as high cost, low central nervous system penetration, and infusion reactions. However, more clinical experience could prevent infusion reactions. The reduced enzyme transportation across the blood–brain barrier decreases central nervous system penetration and limits overall ERT efficacy [20]; however, the low permeability is not a limitation for MPS IVA, since such patients do not exhibit intellectual disabilities. Thus, ERT can lead to beneficial outcomes, such as the prevention of disease progression to some extent. According to a previous study [20], ERT was confirmed to decrease urinary KS (uKS), increase the distance of the 6-min walk test (6MWT), increase climbing stairs in the 3-min stair climb test (3MSCT), increase the scores of the MPS-Health Assessment Questionnaire (HAQ) in self-care, caregiver assistance, and mobility, and increase the level of forced vital capacity (FVC), the first second of forced expiration (FEV1), and maximal voluntary ventilation (MVV).

Morquio A is a skeletal disorder involving multiorgan systems. Elosulfase alfa could effectively improve patient endurance, pulmonary function, and quality of life [20]; however, few systematic reviews exist focusing on the improvement of the aforementioned after ERT and there have been no meta-analyses. Therefore, the impact of elosulfase alfa treatment on key outcomes remains unclear. To evaluate the efficacy of elosulfase alfa for MPS IVA and the symptoms that can potentially be improved by ERT, we used randomized controlled trials (RCTs) with meta-analysis. Non-RCT designs, as part of the inclusion criteria, were also included in the meta-analysis.

## 2. Materials and Methods

We referenced the National Center for Biotechnology Information, U.S. National Library of Medicine National Institutes of Health (PubMed), Excerpta Medica dataBASE (EMBASE), and Cochrane Library databases. The search strategy (MUCOPOLYSACCHARIDOSIS TYPE IVA OR MUCOPOLYSACCHARIDOSIS IVA) AND (ELOSULFASE ALFA OR ENZYME REPLACEMENT THERAPY) was applied to PubMed. An expanded search query with the following keywords was used for EMBASE: mucopolysaccharidosis AND type AND IVA OR “mucopolysaccharidosis”/exp OR mucopolysaccharidosis AND IVA combined with “elosulfase alfa”/exp OR elosulfase alfa OR “enzyme”/exp OR enzyme AND replacement AND (“therapy”/exp OR therapy).

The systematic review was conducted and reported in accordance with the Cochrane Collaboration [21] and PRISMA Statement, respectively. Research that involved RCTs and primary studies without restriction of years was found. Furthermore, all titles and abstracts were screened and full text articles regarding all potentially related studies or trials were obtained. Ultimately, we used pooled analysis of proportions to evaluate four clinical cohort studies and two RCTs (Figure 1). We chose RCTs of elosulfase alfa in patients with a confirmed diagnosis of MPS IVA. If more than five trials meeting these criteria were identified, lower-powered studies (open-label and non-randomized trials, controlled or otherwise, including quasi-experimental designs), as long as the sample size was more than or equal to five, would also be included.

### Proportional Meta-Analysis

A meta-analysis was performed to analyze the studies to evaluate the results of the uKS levels, 6MWT, 3MSCT, MPS-HAQ of self-care, caregiver assistance and FVC, FEV1, and MVV in MPS IVA patients with ERT or placebo (four clinical cohort studies [22,23,24,25] and two RCTs [26,27]) including a total of 550 patients (ERT treatment: 327; placebo treatment: 223). A random-effects model was developed and applied to the two groups of ERT and placebo treatment. The selection criteria for Hughes et al. [22] and Hendriksz et al. [23,25,26] were based on MOR-005 (ClinicalTrials.gov Identifier: NCT01415427). The selection criteria in studies by Hendriksz et al. [24,27] were based on MOR-005 and MOR-004 (ClinicalTrials.gov Identifier: NCT01275066). There were 15 independent reviewers involved in the selection of the included studies.

We calculated the approximate standard error estimates from 95% confidence intervals (CI) by assuming a normal distribution. For the latter analyses, we used the frequentist approach described by Hierarchical Bayes Models [28]. We detected and quantified the statistical heterogeneity via Cochran’s Q test and the I^2^ metric, respectively. Results with *p* < 0.05 were deemed statistically significant. The Comprehensive Meta-Analysis software package was used to conduct this meta-analysis (Version 3).

## 3. Results

Table 1 shows the numbers of patients included in this meta-analysis and their mean ages, mean follow-up years, and genders. All patients were between 10 to 50 years old, and their mean age was 30.0 years. These studies were all conducted in the UK.

Figure 2 shows the proportional meta-analysis result for a pooled proportion from two cohort studies and one RCT for uKS in MPS IVA [22,25,26]. The inconsistency level (I^2^) is 0.00% with no heterogeneity (*p* = 0.872). Thus, the use of a fixed-effects model than a random-effects model is better for this study. The pooled proportions analysis showed that the difference in means of the uKS between the ERT and the placebo treatment groups was −0.260 [95% CI: −0.405, −0.115]. The effect differences favored the ERT treatment group over the placebo treatment group (*p* < 0.001).

Figure 3 reveals the proportional meta-analysis result for a pooled proportion from two cohort studies and two RCTs for the 6MWT (Figure 3a) and 3MSCT (Figure 3b) in MPS IVA [22,25,26,27]. The inconsistency levels (I^2^) are 0.00% and 0.00% (*p* = 0.935 and 0.976) with no heterogeneity. As shown in Figure 2, it is better to use a fixed-effects model than a random-effects model. The difference in means of 6MWT and 3MSCT between the ERT treatment group and the placebo treatment group were −0.102 [95% CI: −0.238, 0.034] and −0.182 [95% CI: −0.333, −0.031]. The effect differences favored the ERT treatment group over the placebo treatment group (*p* = 0.042 and 0.018).

Figure 4 shows the proportional meta-analysis result for a pooled proportion from two cohort studies for MPS-HAQ of self-care (Figure 4a), caregiver assistance (Figure 4b), and mobility (Figure 4c) in MPS IVA [22,23]. The inconsistency levels (I^2^) are 87.80%, 0.00%, and 87.13% (*p* < 0.001, *p* = 0.907 and *p* < 0.001), with high heterogeneity in self-care and mobility and no heterogeneity in caregiver assistance. Using a random-effects model in self-care and mobility and a fixed-effects model in caregiver assistance is deemed better. The pooled proportions analysis showed that the difference in means of self-care, caregiver assistance, and mobility between the ERT and the placebo treatment groups were −0.360 [95% CI: −0.974, 0.253], −0.408 [95% CI: −0.594, −0.222], and −0.587 [95% CI: −1.201, 0.027]. The effect differences favored the ERT treatment group over the placebo treatment group (*p* = 0.049, *p* < 0.001, and *p* = 0.041).

Figure 5 reveals the proportional meta-analysis result for a pooled proportion from two cohort studies and one RCT for FVC (Figure 5a) [22,24,26], FEV1 (Figure 5b) [22,24], and MVV (Figure 5c) [22,24,26,27] in MPS IVA. The inconsistency levels (I^2^) are 0.00%, 18.90%, and 0.00% (*p* = 0.631, *p* = 0.296, and *p* = 0.902) with no heterogeneity. It is better to use a fixed-effects model than a random-effects model. The difference in means of FVC, FEV1, and MVV between the ERT treatment group and the placebo treatment group were −0.293 [95% CI: −0.473, −0.114], −0.311 [95% CI: −0.601, −0.020], and −0.213 [95% CI: −0.378, −0.048]. The effect differences favored the ERT treatment group over the placebo treatment group (*p* = 0.001, *p* = 0.025, and *p* = 0.011).

## 4. Discussion

This meta-analysis is the first to assess the efficacy of IV elosulfase alfa for the treatment of patients with MPS IVA from all age groups. Furthermore, it is the first review to systematically incorporate data from non-RCT studies for this disorder. Our meta-analysis results revealed that elosulfase alfa effectively reduces uKS and improves the performance of 6MWT, 3MSCT, FVC, FEV1, and MVV and the score of MPS-HAQ in self-care, caregiver assistance, and mobility.

Our systematic review results indicate that limited literature has been published regarding MPS IVA due to its rare nature. The limitations encountered in this meta-analysis are the same as that of several rare diseases and conditions found in small populations. Due to the limited number of studies, detection of underlying statistical heterogeneity [29] is considered low with the results of the uKS, 6MWT, 3MSCT, MPS-HAQ of caregiver assistance, FVC, FEV1, and MVV in our study. The hypothesis that heterogeneity is present but remains statistically undetected cannot be ruled out [30]. To solve this problem, Bayesian model and random forest classification procedure were considered to evaluate the accuracy of the classification model [31,32].

Due to progress in technology, novel treatment options are now available, even for rare diseases. However, its widespread use may not be feasible nor be covered by health systems due to a dearth of evidence supporting or quantifying the advantages of these therapies. It is important to use systematic reviews regarding existing expensive treatments for rare diseases to evaluate the clinical decision-making process. An evaluation could also be performed to profile patients who attained a good response to each treatment. Evidence from RCTs may not always be available when elucidating rare diseases. Therefore, it is necessary to support the decision-making process by other designs.

In our study, we included prospective trials to avoid memory and selection bias. This is one of the strengths of our study. Contrary to retrospective trials, the advantage of a prospective trial is that it collected data accordingly with the aimed outcomes. Moreover, we only included case series with n > 5, which is another strength of our study, as it would increase the statistical power of the findings of our meta-analysis.

The effect of ERT on the other outcomes in our study could not be defined as a priori due to the lack of available data and the heterogeneity of the included studies. Elosulfase alfa is effective in treating phenotypes according to the data obtained in this meta-analysis. Nevertheless, it is necessary to evaluate some differences in the response as some studies included did not show data separately for each phenotype.

By increasing lysosomal degradation, uKS could be decreased following elosulfase alfa treatment [33,34,35], and the effect is confirmed by this meta-analysis. uKS can be potentially utilized to differentiate MPS IVA from other forms of MPS; however, because the bone deformation is irreversible, skeletal improvement following ERT for MPS IVA cannot be predicted by the variation in uKS [36].

According to our meta-analysis, patients can improve their performance in 6MWT and 3MSCT after ERT. Nevertheless, surgical procedures were not excluded in the studies. The 6MWT and 3MSCT results can potentially be affected by surgical procedures. We could not evaluate efficacy because of the small sample size and absence of a control group. 6MWT distances in untreated patients decreased according to the Morquio A Clinical Assessment Program natural history study following the progression of untreated Morquio A syndrome over 2 years [37].

For quality of life, improvement in the domains of self-care, mobility, and caregiver assistance over the 5-year study was observed among patients according to our study based on the MPS-HAQ. Furthermore, no negative effects in the quality of life were noted in children with Morquio A syndrome [38], although the quality of life would be expected to decline when the disease progresses.

The effects of ERT response in MPS IVA cannot be currently predicted [39]. Although there are approximately 10% of patients who show an excellent response, a small group of patients does not respond to ERT. Therefore, patients with Morquio A should undergo at least 12 months of elosulfase alfa treatment for their responses to be appropriately evaluated. The assessment tools currently available for determining quality of life are adequate to capture the treatment benefits for patients [40]. Hence, improved and more innovative daily lifestyle measures are necessary to assess the clinical response adequately.

Patients with MPS IVA are at risk of declining pulmonary function as their condition progresses; however, according to our meta-analysis, increases in FVC, FEV1, and MVV were observed. Nevertheless, due to the absence of comparison with an age-matched group of untreated patients, this can be partially attributed to growth [24]. Therefore, the true impact of this treatment on pulmonary function is yet to be determined.

There are some limitations of our study. Because MPS IVA is a rare disease, it is difficult to find studies with an adequate number of patients that could be compared in a meta-analysis. The missing data in some studies limited our sample size enrollment. Another limitation is that there are few studies regarding proinflammatory and prooxidant state in patients with MPS IVA with ERT. According to a study by Donida et al. [41], interleukin-6 and glutathione levels were elevated after ERT in patients with MPS IVA, suggesting a proinflammatory and prooxidant state fostered by this disease. However, patients with MPS IVA who did not receive ERT for comparison were non-existent. This comparison is warranted to optimize patient outcomes according to further investigation of ERT with combined antioxidant and anti-inflammatory agent therapies.

## 5. Conclusions

Our meta-analysis showed that ERT for MPS IVA has beneficial effects on uKS, physical performance, life quality, and respiratory function. Patients with MPS IVA can be started on ERT as soon as the diagnosis is established. Long-term outcome studies are necessary to determine the effect of elosulfase alfa on the patients’ bones. To evaluate the efficacy of ERT treatment, we should moderate the long-term progression of MPS IVA.

## Figures and Tables

**Figure 1 jpm-12-01338-f001:**
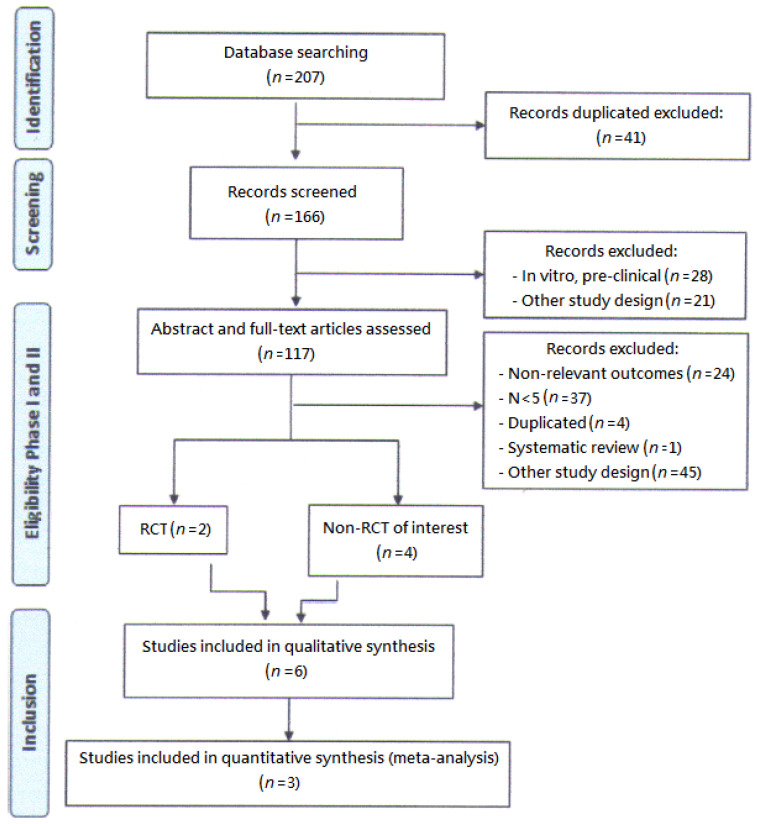
Elosulfase alfa for MPS IVA: A flowchart of a systematic review.

**Figure 2 jpm-12-01338-f002:**
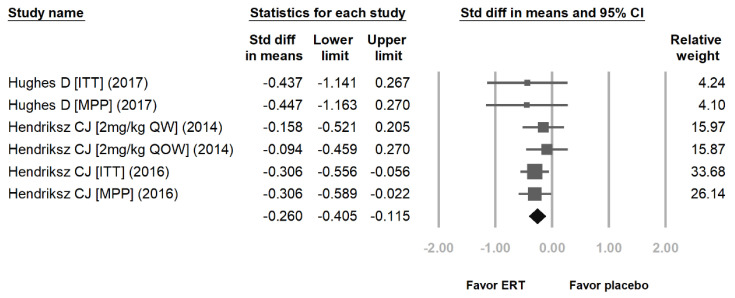
Proportional meta-analysis result for a pooled proportion from two cohort studies and one randomized controlled trial (RCT) for urinary keratan sulfate (uKS) in MPS IVA. The pooled proportions analysis showed that the difference in means of uKS between the enzyme replacement therapy (ERT) group and the placebo treatment group was −0.260 [95% CI: −0.405, −0.115; heterogeneity: I^2^ = 0.00%, *p* = 0.872; overall effect: *p* < 0.001] (ITT: intention to treat; MPP: modified per protocol) [22,25,26]. (Reprinted/adapted with permission from Ref. [22], 2017, Hughes et al.; Ref. [25], 2016, Hendriksz et al. and Ref. [26], 2014, Hendriksz et al.)

**Figure 3 jpm-12-01338-f003:**
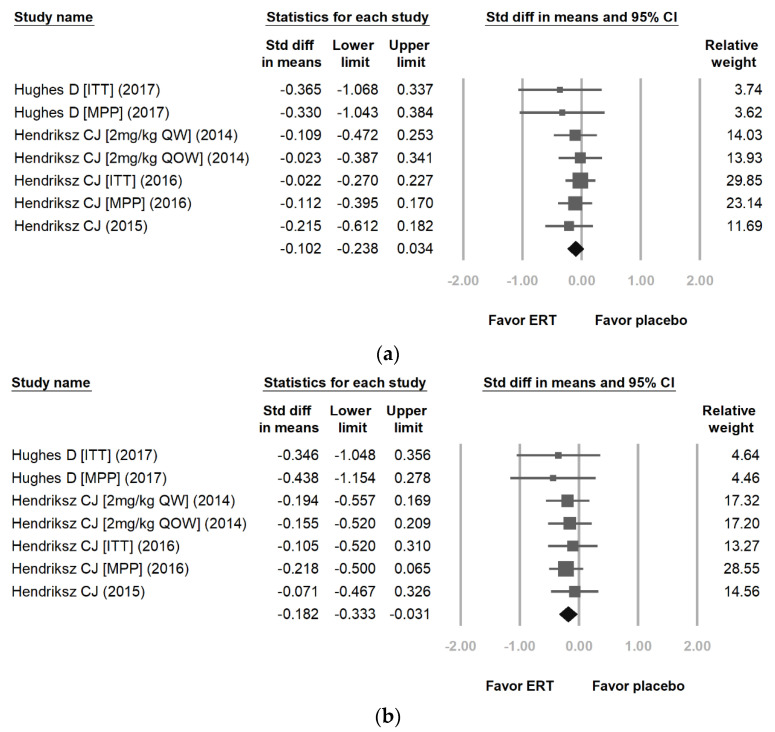
Proportional meta-analysis result for a pooled proportion from two cohort studies and two randomized controlled trials (RCTs) for the (**a**) 6-min walk test (6MWT) and (**b**) 3-min stair climb test (3MSCT) in MPS IVA. The pooled proportions analysis showed that the difference in means of 6MWT and 3MSCT between the enzyme replacement therapy (ERT) group and the placebo treatment group were (**a**) −0.102 [95% CI: −0.238, 0.034; heterogeneity: I^2^ = 0.00%, *p* = 0.935; overall effect: *p* = 0.042] and (**b**) −0.182 [95% CI: −0.333, −0.031; heterogeneity: I^2^ = 0.00%, *p* = 0.976; overall effect, *p* = 0.018] (ITT: intention to treat; MPP: modified per protocol) [22,25,26,27]. (Reprinted/adapted with permission from Ref. [22], 2017, Hughes et al.; Ref. [25], 2016, Hendriksz et al.; Ref. [26], 2014, Hendriksz et al. and Ref. [27], 2015, Hendriksz et al.)

**Figure 4 jpm-12-01338-f004:**
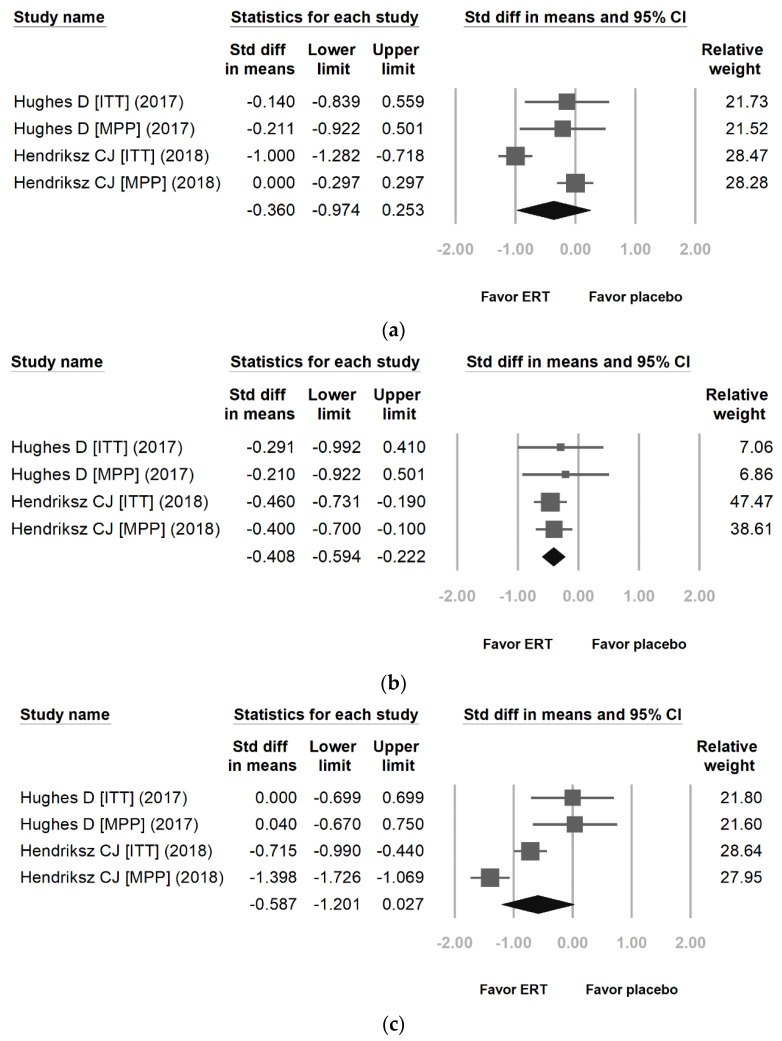
Proportional meta-analysis result for a pooled proportion from two cohort studies for MPS-Health Assessment Questionnaire (HAQ) of (**a**) self-care, (**b**) caregiver assistance, and (**c**) mobility in MPS IVA. The pooled proportions analysis showed that the difference in means of self-care, caregiver assistance, and mobility between the enzyme replacement therapy (ERT) group and the placebo treatment group were (**a**) −0.360 [95% CI: −0.974, 0.253; heterogeneity: I^2^ = 87.08%, *p* < 0.001; overall effect: *p* = 0.049], (**b**) −0.408 [95% CI: −0.594, −0.222; I^2^ = 0.00%, *p* = 0.907; overall effect: *p* < 0.001], and (**c**) −0.587 [95% CI: −1.201, 0.027; I^2^ = 87.13%, *p* < 0.001; overall effect: *p* = 0.041] (ITT: intention to treat; MPP: modified per protocol) [22,23]. (Reprinted/adapted with permission from Ref. [22], 2017, Hughes et al. and Ref. [23], 2018, Hendriksz et al.)

**Figure 5 jpm-12-01338-f005:**
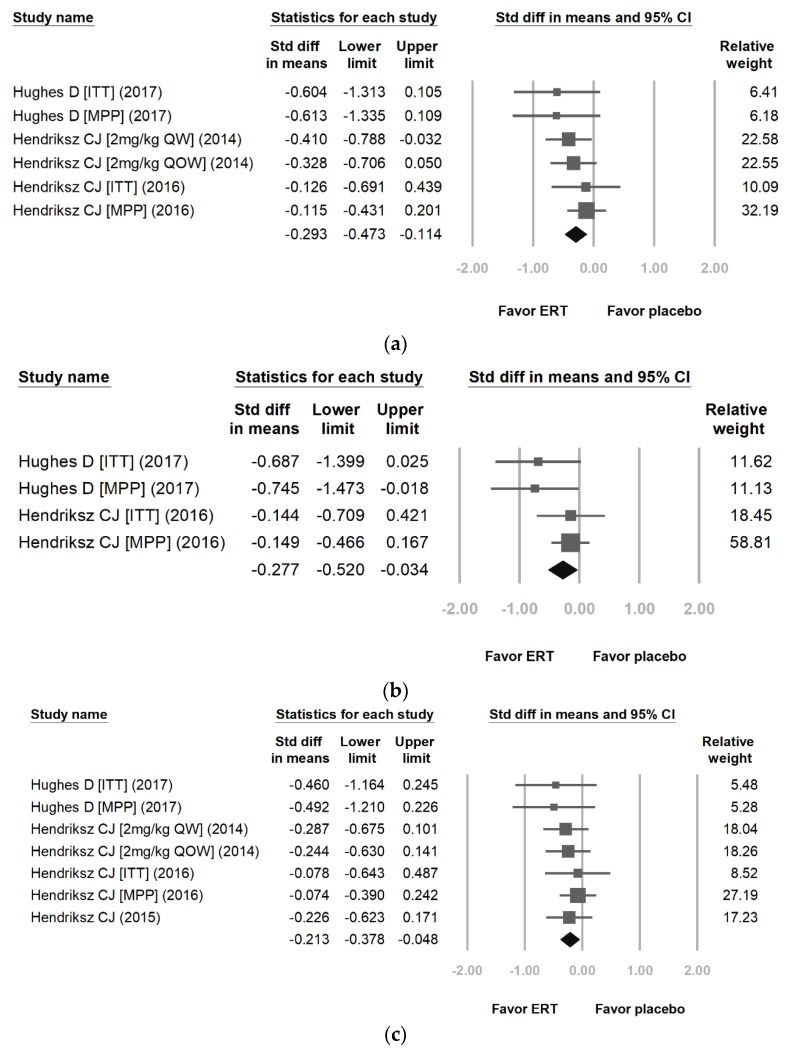
Proportional meta-analysis result for a pooled proportion from two cohort studies and one randomized controlled trial (RCT) for forced vital capacity (FVC), the first second of forced expiration (FEV1), and maximal voluntary ventilation (MVV) in MPS IVA. The pooled proportions analysis showed that the difference in means of (**a**) FVC [22,24,26], (**b**) FEV1 [22,24], and (**c**) MVV [22,24,26,27] between the enzyme replacement therapy (ERT) group and the placebo treatment group were (**a**) −0.293 [95% CI: −0.473, −0.114; heterogeneity: I^2^ = 0.00%, *p* = 0.631; overall effect: *p* = 0.001], (**b**) −0.311 [95% CI: −0.601, −0.020; heterogeneity: I^2^ = 18.90%, *p* = 0.296; overall effect: *p* = 0.025], and (**c**) −0.213 [95% CI: −0.378, −0.048; heterogeneity: I^2^ = 0.00%, *p* = 0.902; overall effect: *p* = 0.011] (ITT: intention to treat; MPP: modified per protocol). (Reprinted/adapted with permission from Ref. [22], 2017, Hughes et al.; Ref. [24], 2016, Hendriksz et al.; Ref. [26], 2014, Hendriksz et al. and Ref. [27], 2015, Hendriksz et al.).

**Table 1 jpm-12-01338-t001:** Basic patient and study characteristics. Enzyme replacement therapy (ERT).

	ERT	Placebo	Total
Number of patients	327	223	550
Mean age (years)	30.9	26.9	30.0
Mean follow-up (years)	2.3	1.4	1.9
Gender (male # of percentage)	115 (35.1%)	22 (10.0%)	137 (24.9%)

## Data Availability

All data are present within the article.

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
