# Peer review of "Efficacy of Intravenous Elosulfase Alfa for Mucopolysaccharidosis Type IVA: A Systematic Review and Meta-Analysis"

_jpm, 2022, doi:10.3390/jpm12081338_

Round 1

Reviewer 1 Report

Introduction

The Introduction lacks a description of the outcomes that were evaluated by the meta-analysis: urinary keratan sulfate levels, 6-min walk test, 3-min stair climb test, MPS-Health Assessment Questionnaire of self-care, caregiver assistance and mobility, forced vital capacity, the first second of forced expiration, and maximal voluntary ventilation. Primary and secondary outcomes were not defined. It is Ok to place this kind of information in Materials and Methods, but the reader must understand the relevance of those outcomes to mobility, pulmonary function, and quality of life. The relevance of reduction of urinary keratan sulfate levels is the least clear - in the Discussion it is stated that "[we] demonstrate pharmacodynamic effects of ERT treatment (246)" - pharmacodynamic effects are relevant clinical outcomes? No outcome was chosen to evaluate improvement of bone lesions.  

Materials and Methods

The methodology of the systematic review and meta-analysis was extremely simplified by a citation: “The systematic review was conducted as proposed by the Cochrane Collaboration [21]”. 103

As systematic reviews based on Cochrane Collaboration vary enormously, it is important that the authors state clearly: (1) the number of independent evaluators that selected the RCTs and cohort studies, the agreement among them and the methodology to solve disagreement; (2) the scales or checklists they used to evaluate article quality; (3) assessment of risk of bias in included studies; (4) assessment of reporting bias.

The authors cite Higgins et al. (2009) as the basis for the frequentist approach they described. Nevertheless, Higgins et al. (2009) criticized this approach and proposed a hierarchical Bayesian model for meta-analysis.

“We detected and quantified the statistical heterogeneity via the I2 metric, respectively”. 122/123 – What was the other metric used to detect and quantify heterogeneity?

Results

The results section should initiate with a description of the included studies. I had difficulties of finding them in the references, e.g., there are 3 Hendriksz CJ et al. 2014 articles, and 2 Hendriksz CJ et al. 2016 articles.

The meta-analysis included four cohort studies and two randomized controlled trials; however, they referred the two groups of comparison as ERT and placebo. I was wondering what a placebo group in a cohort study is. However, as there is no description of the included studies, I could not retrieve the cohort studies and check what were the comparison groups.

Of the 9 outcomes chosen by the authors, 6 presented the heterogeneity statistic I2 equal to zero. Although in the Discussion section they mention that this statistic can be biased in small meta-analyses, they did not in the Results section present any measure of I2 uncertainty, like the confidence interval of I2 – not to be confounded with the confidence interval of the difference between the two groups – ERT and placebo.

Discussion

According to a study by Donida et al. in a cohort of MPS IVA patients receiving ERT therapy [39], interleukin-6 and decreased glutathione levels were elevated. 278-279 Unclear sentence, please rewrite it.

This is considered as a major limitation of the present meta-analysis if the use of graphs and tests for the assessment of funnel plot asymmetry was avoided. 220-221 Unclear sentence, please rewrite it.

The effect of ERT to other study outcomes could not be defined as a priori due to the lack of available data and to the heterogeneity of the included studies. 239-240 Another unclear sentence, please rewrite it.

Elosulfase alfa is effective in treating phenotypes according to the data obtained in this meta-analysis. 240-241 No definition of phenotype is mentioned in the Introduction and Results. I encourage the authors to form subgroups according to the clinical presentation of MPS IVA – classical, nonclassical, and intermediate forms. But if this is not possible the reference to phenotypes should be excluded.

There are some limitations in our study. First, our study is a retrospective study. Because MPS IVA is a rare disease, it is difficult to find an adequate number of patients for the study; moreover, conducting a long-term follow-up study is not feasible. 273-275 As the study is a systematic review and meta-analysis, it could not be a prospective study, neither a retrospective study, as a matter of fact – this classification cannot be applied to systematic reviews, but to cohort studies. The difficulty is not to find an adequate number of patients for the study, but to find studies with an adequate number of patients that could be compared in a meta-analysis. The difference may be subtle, but it is important as to do a meta-analysis, one does not recruit patients – this role is reserved for RCT and cohort studies.

According to a study by Donida et al. in a cohort of MPS IVA patients receiving ERT therapy [39], interleukin-6 and decreased glutathione levels were elevated. 278-279 Unclear sentence, please rewrite it.

Author Response

Introduction

1. The Introduction lacks a description of the outcomes that were evaluated by the meta-analysis: urinary keratan sulfate levels, 6-min walk test, 3-min stair climb test, MPS-Health Assessment Questionnaire of self-care, caregiver assistance and mobility, forced vital capacity, the first second of forced expiration, and maximal voluntary ventilation. Primary and secondary outcomes were not defined. It is Ok to place this kind of information in Materials and Methods, but the reader must understand the relevance of those outcomes to mobility, pulmonary function, and quality of life.

=> Thank you for your suggestion. We will add the description of the outcomes that were evaluated by the meta-analysis. “ According to previous study [20], ERT was confirmed to decrease urinary keratan sul-fate (uKS), increase the distance of 6-min walk test (6MWT), increase climbing stairs of 3-min stair climb test (3MSCT), increase scores of MPS-Health Assessment Question-naire (HAQ) in self-care, caregiver assistance and mobility, increase the level of forced vital capacity (FVC), the first second of forced expiration (FEV1), and maximal volun-tary ventilation (MVV).” in Line 84.

2. The relevance of reduction of urinary keratan sulfate levels is the least clear - in the Discussion it is stated that "[we] demonstrate pharmacodynamic effects of ERT treatment (246)" - pharmacodynamic effects are relevant clinical outcomes? No outcome was chosen to evaluate improvement of bone lesions.

=> Thank you for your suggestion. The pharmacodynamic effects could affect the decrease rate of uKS;  however, bone lesions could not decrease because the bone deformation is irreversible. We will describe it in Line 257.

Materials and Methods

1. The methodology of the systematic review and meta-analysis was extremely simplified by a citation: “The systematic review was conducted as proposed by the Cochrane Collaboration [21]”. As systematic reviews based on Cochrane Collaboration vary enormously, it is important that the authors state clearly: (1) the number of independent evaluators that selected the RCTs and cohort studies, the agreement among them and the methodology to solve disagreement; (2) the scales or checklists they used to evaluate article quality; (3) assessment of risk of bias in included studies; (4) assessment of reporting bias.

=> Thank you for your suggestion. We will modify the statement. “ The systematic review was conducted as proposed by the Cochrane Collaboration [21] and reported as proposed by PRISMA Statement.” in Line 110. And “ We choose RCTs of elosulfase alfa ERT for treatment of patients with a confirmed di-agnosis of MPS IVA. If < five trials meeting these criteria were identified, low-er-powered studies (open-label and non-randomized trials, controlled or otherwise, including quasi-experimental designs), as long as the sample size was ≥ five, would also be included.” in Line 115.

2. The authors cite Higgins et al. (2009) as the basis for the frequentist approach they described. Nevertheless, Higgins et al. (2009) criticized this approach and proposed a hierarchical Bayesian model for meta-analysis.

=> Thank you for your suggestion. We will modify the statement. “ For the latter analyses, we used the frequentist approach described by Hierarchical Bayes Models [28].” in Line 135.

3. “We detected and quantified the statistical heterogeneity via the I2 metric, respectively”. 122/123 – What was the other metric used to detect and quantify heterogeneity?

=> Cochran's Q test was the other metric used to detect and quantify heterogeneity.

Results

1. The results section should initiate with a description of the included studies. I had difficulties of finding them in the references, e.g., there are 3 Hendriksz CJ et al. 2014 articles, and 2 Hendriksz CJ et al. 2016 articles.

=> Thank you for your suggestion. We will add the references in figures and results part.

2. The meta-analysis included four cohort studies and two randomized controlled trials; however, they referred the two groups of comparison as ERT and placebo. I was wondering what a placebo group in a cohort study is. However, as there is no description of the included studies, I could not retrieve the cohort studies and check what were the comparison groups.

=> Patients with Morquio A aged ≥5 years were randomised (1:1:1) to receive elosulfase alfa 2.0 mg/kg/every other week (qow), elosulfase alfa 2.0 mg/kg/week (weekly) or placebo for 24 weeks.

3. Of the 9 outcomes chosen by the authors, 6 presented the heterogeneity statistic I2 equal to zero. Although in the Discussion section they mention that this statistic can be biased in small meta-analyses, they did not in the Results section present any measure of I2 uncertainty, like the confidence interval of I2 – not to be confounded with the confidence interval of the difference between the two groups – ERT and placebo.

=> Thank you for your suggestion. We will add the p-value of I2 in results and figures. The confidence interval of I2 could not be calculated in the Comprehensive Meta-Analysis software package (Version 3).

Discussion

1. According to a study by Donida et al. in a cohort of MPS IVA patients receiving ERT therapy [39], interleukin-6 and decreased glutathione levels were elevated. 278-279 Unclear sentence, please rewrite it.

=> Thank you for your suggestion. We will rewrite this sentence. “ According to a study by Donida et al. [39], interleukin-6 and glutathione levels were elevated after ERT in MPS IVA patients.” in Line 292.

2. This is considered as a major limitation of the present meta-analysis if the use of graphs and tests for the assessment of funnel plot asymmetry was avoided. 220-221 Unclear sentence, please rewrite it.

=> Thank you for your suggestion. We will rewrite this sentence. “ This can be the major limitation of the present meta-analysis if we avoid the use of graphs and tests for the assessment of funnel plot asymmetry.” in Line 234.

3. The effect of ERT to other study outcomes could not be defined as a priori due to the lack of available data and to the heterogeneity of the included studies. 239-240 Another unclear sentence, please rewrite it.

=> Thank you for your suggestion. We will rewrite this sentence. “ The effect of ERT to other outcomes in our study could not be defined as a priori due to the lack of available data and the heterogeneity of the included studies” in Line 253.

4. Elosulfase alfa is effective in treating phenotypes according to the data obtained in this meta-analysis. 240-241 No definition of phenotype is mentioned in the Introduction and Results. I encourage the authors to form subgroups according to the clinical presentation of MPS IVA – classical, nonclassical, and intermediate forms. But if this is not possible the reference to phenotypes should be excluded.

=> Thank you for your suggestion. It is  inconvenient to form subgroups according to the clinical presentation of MPS IVA. We will exclude the reference to phenotypes and form subgroups according to the clinical presentation of MPS IVA in next study.

5. There are some limitations in our study. First, our study is a retrospective study. Because MPS IVA is a rare disease, it is difficult to find an adequate number of patients for the study; moreover, conducting a long-term follow-up study is not feasible. 273-275 As the study is a systematic review and meta-analysis, it could not be a prospective study, neither a retrospective study, as a matter of fact – this classification cannot be applied to systematic reviews, but to cohort studies. The difficulty is not to find an adequate number of patients for the study, but to find studies with an adequate number of patients that could be compared in a meta-analysis. The difference may be subtle, but it is important as to do a meta-analysis, one does not recruit patients – this role is reserved for RCT and cohort studies.

=> Thank you for your suggestion. We will modify this mention. “ Because MPS IVA is a rare disease, it is difficult to find studies with an adequate number of patients that could be compared in a meta-analysis.” in Line 287.

6. 

Reviewer 2 Report

In this study, the authors have aimed to evaluate ERT efficacy with elosulfase alfa in MPS IVA using randomized controlled trials with meta-analysis. A total of 550 patients with 327 ERT treatments and 223 placeboes were included. The study indicates that there are beneficial effects on urinary keratan sulfate, physical performance,  quality of life, and respiratory function improvement with ERT in MPS IVA patients. Because MPS IVA is a rare disease, the meta-analysis study and presentation of the findings are valuable and may interest the reader. The study was straightforward, and the conclusion was consistent with the literature. 

Thanks for your hard work.

Best regards.

Author Response

In this study, the authors have aimed to evaluate ERT efficacy with elosulfase alfa in MPS IVA using randomized controlled trials with meta-analysis. A total of 550 patients with 327 ERT treatments and 223 placeboes were included. The study indicates that there are beneficial effects on urinary keratan sulfate, physical performance,  quality of life, and respiratory function improvement with ERT in MPS IVA patients. Because MPS IVA is a rare disease, the meta-analysis study and presentation of the findings are valuable and may interest the reader. The study was straightforward, and the conclusion was consistent with the literature. => Thanks for your advice and encouragement. We will make an all-out effort in our studies.

Reviewer 3 Report

This manuscript is very well written. I only have a few suggestions/questions for the authors:

- Please italicize GALNS gene throughout the paper. 

- Review the text carefully as there are several words with wrong hyphens;

-Line 53: please remove "such as" for clarity. 

-Lines 81-81 could be rephrased for clarity, such as: "The low enzyme penetration through the blood-brain barrier reduces central nervous system penetration and limits the efficacy of all ERTs; however, the low permeability is not a limitation for MPS IVA, as patients do not have intellectual disabilities. ERT can thus provide beneficial outcomes, including preventing disease progression to a certain extent”.

Line 127 -  change "sexes" for "gender".

Points I think should be included somewhere in the text:

1) What is the main question for the meta-analysis? What are the primary and secondary objectives?

2) The inclusion and exclusion criteria for each one of the included studies should be presented.

3) How many independent reviewers were involved in the selection of the included studies?

Author Response

1.Please italicize GALNS gene throughout the paper.

=> Thank you for your suggestion. We will  italicize GALNS gene throughout our manuscript as highlighted in yellow.

2. Line 53: please remove "such as" for clarity.

=> Thank you for your suggestion. We will remove "such as" for clarity.

3. Lines 81-81: could be rephrased for clarity, such as: "The low enzyme penetration through the blood-brain barrier reduces central nervous system penetration and limits the efficacy of all ERTs; however, the low permeability is not a limitation for MPS IVA, as patients do not have intellectual disabilities. ERT can thus provide beneficial outcomes, including preventing disease progression to a certain extent”.

=> Thank you for your suggestion. We will modify the sentence as highlighted in yellow in Line 80.

4. Line 127: change "sexes" for "gender".

=> Thank you for your suggestion. We will  change "sexes" for "gender" as highlighted in yellow in Line 133.

5. Points I think should be included somewhere in the text:

(1) What is the main question for the meta-analysis? What are the primary and secondary objectives?

(2) The inclusion and exclusion criteria for each one of the included studies should be presented.

(3) How many independent reviewers were involved in the selection of the included studies?

=> Thank you for your suggestion.

(1) Main question: What is the impact of elosulfase alfa treatment on key outcomes ?

Primary objective: To evaluate the efficacy of elosulfase alfa for MPS IVA.

Secondary objective: Which symptoms could be improved by  elosulfase alfa.

They are highlighted in yellow in Line 88.

(2) The selection criteria for Hughes et al. [22] and Hendriksz et al. [23,25,26] were based on MOR-005 (ClinicalTrials.gov Identifier: NCT01415427). Studies from Hendriksz et al. [24,27] had selection criteria based on MOR-005 and MOR-004 (ClinicalTrials.gov Iden-tifier: NCT01275066).

They are highlighted in yellow in Line 120.

(3) There were fifteen independent reviewers involved in the selection of the included studies.

They are highlighted in yellow in Line 124.

Round 2

Reviewer 1 Report

“Among previous studies, there was no meta-analysis. Therefore, it is still unclear about the impact of elosulfase alfa treatment on key outcomes. To evaluate the efficacy of elosulfase alfa for MPS IVA and which symptoms could be improved by ERT, we used randomized controlled trials (RCTs) with meta-analysis. In other expanded studies, non-RCT designs as part of the inclusion criteria were also included in the meta-analysis.” 94-97

The authors contrast their meta-analysis with previous (it is not clear) “expanded studies” (systematic reviews with no meta-analysis?) that included non-RCT [study] designs. Nevertheless, their meta-analysis included RCTs and cohort studies. Please make the idea of this paragraph clearer and improve English language stylistic.

“We choose RCTs[…]” 115

Verb tense not adequate for Materials and Methods.

“We detected and quantified the statistical heterogeneity via the I2 metric, respectively”. 136-137

According to the authors’ answer, it should be:

“We detected and quantified the statistical heterogeneity via Cochran's Q test and the I2 metric, respectively”. 

“The inconsistency level (I2) is 0.00% with no heterogeneity (= 0.872, [95% CI: −0.405, −0.115])”. 147-148

It is Ok for the authors to present only the p value of I2 as the software they used cannot calculate the CI of I2. So, the CI they placed after the p value corresponds to the difference in means of the uKS between the ERT and the placebo treatment groups.

Version 1: “This is considered as a major limitation of the present meta-analysis if the use of graphs and tests for the assessment of funnel plot asymmetry was avoided”. 

Version 2: “This can be the major limitation of the present meta-analysis if we avoid the use of graphs and tests for the assessment of funnel plot asymmetry.” 234-236

It is not clear the idea that the authors wanted to communicate with this sentence, as there can be an English language stylistic flaw here. Nevertheless, funnel plot asymmetry, be it visually examined or evaluated by more formal statistical methods, has a very limited power to detect bias if the number of studies is small as in the present meta-analysis. The limitations encountered by this meta-analysis are shared with several rare diseases and conditions found in small populations. Maybe the authors could briefly mention efforts to overcome these limitations – see the following publications:

Hampson LV, Whitehead J, Eleftheriou D, Brogan P 2014. Bayesian methods for the design and interpretation of clinical trials in very rare diseases. Statistics in Medicine 33: 4186–4201. DOI:10.1002/sim.6225.

Speiser JL, Durkalski VJ, Lee WM 2015. Random forest classification of etiologies for an orphan disease. Statistics in Medicine 34: 887–899. DOI:10.1002/sim.6351.

Author Response

  1. “Among previous studies, there was no meta-analysis. Therefore, it is still unclear about the impact of elosulfase alfa treatment on key outcomes. To evaluate the efficacy of elosulfase alfa for MPS IVA and which symptoms could be improved by ERT, we used randomized controlled trials (RCTs) with meta-analysis. In other expanded studies, non-RCT designs as part of the inclusion criteria were also included in the meta-analysis.” 94-97

The authors contrast their meta-analysis with previous (it is not clear) “expanded studies” (systematic reviews with no meta-analysis?) that included non-RCT [study] designs. Nevertheless, their meta-analysis included RCTs and cohort studies. Please make the idea of this paragraph clearer and improve English language stylistic.

=> Thank you for your suggestion. We will modify our paragraph as below. “ To evaluate the efficacy of elosulfase alfa for MPS IVA and the symptoms that can potentially be improved by ERT, we used randomized controlled trials (RCTs) with me-ta-analysis. Non-RCT designs, as part of the inclusion criteria, were also included in the meta-analysis.” In Line 97-100.

2.“We choose RCTs[…]” 115

Verb tense not adequate for Materials and Methods.

=> Thank you for your suggestion. We will modify the verb tense. “ We chose RCTs of elosulfase alfa in patients with a confirmed diagnosis of MPS IVA ” In Line 116-117.

3.“We detected and quantified the statistical heterogeneity via the I2 metric, respectively”. 136-137

According to the authors’ answer, it should be:

“We detected and quantified the statistical heterogeneity via Cochran's Q test and the I2 metric, respectively”.

=> Thank you for your suggestion. We will modify the paragraph. “ We detected and quantified the statistical heterogeneity via Cochran's Q test and the I2 metric, respectively.” In Line 137-138.

  1. “The inconsistency level (I2) is 0.00% with no heterogeneity (p = 0.872, [95% CI: −0.405, −0.115])”. 147-148

It is Ok for the authors to present only the p value of I2 as the software they used cannot calculate the CI of I2. So, the CI they placed after the p value corresponds to the difference in means of the uKS between the ERT and the placebo treatment groups.

=> Thank you for your suggestion. The CI is  the difference in means of the uKS between the ERT and the placebo treatment groups. We will remove the CI in Line 148.

  1. Version 1: “This is considered as a major limitation of the present meta-analysis if the use of graphs and tests for the assessment of funnel plot asymmetry was avoided”.

Version 2: “This can be the major limitation of the present meta-analysis if we avoid the use of graphs and tests for the assessment of funnel plot asymmetry.” 234-236

It is not clear the idea that the authors wanted to communicate with this sentence, as there can be an English language stylistic flaw here. Nevertheless, funnel plot asymmetry, be it visually examined or evaluated by more formal statistical methods, has a very limited power to detect bias if the number of studies is small as in the present meta-analysis. The limitations encountered by this meta-analysis are shared with several rare diseases and conditions found in small populations. Maybe the authors could briefly mention efforts to overcome these limitations – see the following publications:

Hampson LV, Whitehead J, Eleftheriou D, Brogan P 2014. Bayesian methods for the design and interpretation of clinical trials in very rare diseases. Statistics in Medicine 33: 4186–4201. DOI:10.1002/sim.6225.

Speiser JL, Durkalski VJ, Lee WM 2015. Random forest classification of etiologies for an orphan disease. Statistics in Medicine 34: 887–899. DOI:10.1002/sim.6351.

=> Thank you for your suggestion. We will modify the paragraph.

“ The limitations encountered in this meta-analysis are the same as that of several rare diseases and conditions found in small populations.” In Line 236-238.

“ To solve this problem, Bayesian model and random forest classification procedure were considered to evaluate the accuracy of the classification model. [31–32]” In Line 242-243.